# Digital Cultural Heritage Preservation in Art Painting: A Surface Roughness Approach to the Brush Strokes

**DOI:** 10.3390/s20216269

**Published:** 2020-11-03

**Authors:** Anna Mironova, Frederic Robache, Raphael Deltombe, Robin Guibert, Ludovic Nys, Maxence Bigerelle

**Affiliations:** 1CNRS UMR 8201–LAMIH–Laboratoire d’Automatique, de Mécanique et d’Informatique Industrielles et Humaines, University Polytechnique Hauts-de-France, F-59313 Valenciennes, France; anna.mironova@uphf.fr (A.M.); frederic.robache@uphf.fr (F.R.); raphael.deltombe@uphf.fr (R.D.); robin.guibert@uphf.fr (R.G.); 2CRISS-Laboratoire “Centre de Recherche Interdisciplinaire en Sciences de la Société”, Bâtiment des Tertiales, University Polytechnique Hauts-de-France, Rue des Cent Têtes, F-59300 Valenciennes, France; Ludovic.Nys@uphf.fr

**Keywords:** cultural heritage, surface roughness, art painting, digitization

## Abstract

There is a growing interest in cultural heritage preservation. The notion of HyperHeritage highlights the creation of new means of communication for the perception and data processing in cultural heritage. This article presents the Digital Surface HyperHeritage approach, an academic project to identify the topography of art painting surfaces at the scale at which the elementary information of sensorial rendering is contained. High-resolution roughness and imaging measurement tools are then required. The high-resolution digital model of painted surfaces provides a solid foundation for artwork-related information and is a source of many potential opportunities in the fields of identification, conservation, and restoration. It can facilitate the determination of the operations used by the artist in the creative process and allow art historians to define, for instance, the meaning, provenance, or authorship of a masterpiece. The Digital Surface HyperHeritage approach also includes the development of a database for archiving and sharing the topographic signature of a painting.

## 1. Introduction

Digital technologies breathe new life into cultural heritage. Two of the major issues are the preservation of cultural heritage through an optimal digitization of the object itself—and consequently, to allow a collective perception—and the establishment of cultural networks. In the field of painting, digitization focuses on the conservation of the object through the interaction of light on the surface, and reproduces it as faithfully as possible. The digitization of data to preserve the painter’s characteristic modus operandi is another aspect of cultural heritage preservation. The approach, described in this paper, is based on the digital preservation of sensorial perception, and expression of the painter’s thought, creative process, and imprint.

### 1.1. The Digital Surface HyperHeritage: An HyperHeritage Concept

From a scientific perspective, the painting process involves four main domains: colorimetry, surface topography, matter–matter interaction, and tool–matter interaction. These four domains are currently studied in scientific laboratories (for identification, falsification, matter integrity, etc.). At present, however, the analysis tools and methods used are yet to be introduced into the research as techniques for digitization and preservation of the art heritage. The term “HyperHeritage” refers to all hybrid cultural heritage environments augmented with digital information, which invites us to explore new ways to perceive, experience, and practice cultural heritage [1]. In the opinion of the authors, our approach represents a complementary perspective to this concept, and provides a kind of DNA cartography of artistic objects. By extension, this approach can be named Technical HyperHeritage. 

A “Digital Surface HyperHeritage” approach was undertaken as part of an academic project aiming at the identification of the scales in surface topography that contain all of its functional components (from large- to small-sized surface elements). In practice, these scales are generally less than one millimeter. Thus, topography must often be measured at sub-micrometric scales because the optical scattering of brush strokes on a painting can only be deduced from a surface that provides discretization below the micrometer scale. 

### 1.2. The Implementation of the Digital Surface HyperHeritage Approach

The implementation of the Digital Surface HyperHeritage approach involves many technical problems of measurement, archiving, and optimization, and can be summarized in seven steps: Select Instrument, Measure, Characterize, Archive, Visualize, Integrate, Treat (Figure 1): 

**(a)** 
**Define the optimal methodology of automated measurements suited to the topography of the investigated surfaces (apparatus selection and protocol) allowing a set of repetitive, reliable, and robust measurements to be performed and quantified by a quality indicator.**


Many techniques exist to measure the topography of a surface, thus creating differences in the representation of the same surface [2]. Each technique depends on the measured region dimensions, the desired precision of both the measured region and topography amplitude [3], measurement time [4], non-destructive aspects of the tools (plastic deformation due to contact [5]), the topography itself (roughness amplitude and slope [6]), and physical properties of the material (transparency [7], multi-phase [3], color [8]). Even if standards correctly describe the operation of measuring devices [9], each apparatus is separately treated (contact (stylus) instruments [10], confocal chromatic probe [11], phase shifting interferometric microscopy [12], coherence scanning interferometry [13], point autofocus [14], variable focus [15]).

Structure light projection-based methods [16] are considered by standards of surface finishes, as shape measurement technologies rather than as roughness measurement technologies. Indeed, it is uncommon to obtain a lateral resolution of more than 10 µm and a vertical resolution (z-axis) of more than 3 µm [17]. The same observations can be made for laser triangulation [18] for which resolutions, although more precise than fringe projection methods, remain relatively low. However, they allow the measurement of extremely rough topography, which makes it an advantageous technique for highly fractal surfaces or for surfaces with a strong motif variation [18]). In fact, to compare measurement devices on the same surface, it is relevant to acquire the topography and perform a multi-scale decomposition so that the connections on all scale overlays are analyzed [19,20].

The application of surface topography in the field of fine art paintings is not widely used currently. A 2019 review by Elkhuizen et al. of state-of-the-art approaches [21] gathers only 13 publications with a complete metrological analysis of resolutions for three metrological techniques: laser triangulation, structured light projection, and focus variation microscopy. Elkhuizen et al. compare three 3D scanning techniques that were used to capture the surface topography of *Girl with a Pearl Earring* by Johannes Vermeer (c. 1665), a painting in the collection of the Mauritshuis, the Hague [22]. These three techniques are multi-scale optical coherence tomography [23], 3D scanning based on fringe-encoded stereo imaging (at two resolutions) [24], and 3D digital microscopy. The study finds that 3D digital microscopy (focus variation) and multi-scale optical coherence tomography offer the highest measurement accuracy and precision.

**(b)** 
**Deployment of a multi-scale measurement strategy to capture all of the features and particularities of the masterpiece. Indeed, the whole measurement of the masterpiece is technically impossible, at the moment, with a lateral resolution of the order of 1 μm. A three-dimensional multi-resolution and multi-technique approach must then be built.**


The “Holy Grail” in the field of surface topography would be to achieve multi-sensor acquisition, thus allowing measurement over an extended range of amplitudes of roughness with nanometric precision along vertical and lateral resolutions. Scanning all of the relevant scales described in the cultural heritage would be possible, with the help of imaging and spectroscopy-based analytical methods (see review of different scales of characterization by Crina Anca Sandu et al. [25]). Given the current state of knowledge, the design of such instrumentation appears unattainable. Several considerations lead to this conclusion. First, sub-nanometric measurements appear currently to be reserved to scanning probe microscopy (SPM) techniques such as scanning tunneling microscopy (STM) and atomic force microscopy (AFM) [26]. Even in optimized configurations, such as fast scanning AFM, the measurements cannot exceed a scanning area of 1 mm² (without stitching). Ahmad et al. [27] obtain a resolution of 0.5 × 0.2 mm within a single scan with a 0.2 nm resolution in the vertical direction. For example, to achieve a 9.8 nm resolution for a 600 × 125 µm scanning area, an image size of 12,800 × 51,200 pixels would be required. At 20 lines/s scanning speed, the whole area would be processed within 640 s (10.7 min) and the total data amount would be 40,000 megabits (5 GB). If we suppose that it is possible to acquire, with the help of stitching, the surface of a painting with an area of 1 m² at nanometer resolution (which appears technically impossible), the number of stitches required would be about 10^16^, so 5 × 10^16^ GB (Kolos, the largest data center in the world located in the arctic region of Norway, in 2020 offers a storage capacity of 47 ZB (4.7 × 10^13^ GB), i.e., 1000 times less than the storage required for the topography of the whole painting). Regarding interferometer resolutions, the NewView 7300, (ZYGO, Fremont, CA, USA) equipped with a Zygo 100× objective yields a 0.109 µm lateral resolution (and near 0.1 nm depth direction); height amplitude encoded on 4 bytes would require storage of 336 TB. In addition to this memory storage, the measurement time of 10 s for each elementary topography on this interferometer would require 87 years to complete the acquisition, because 2.7 × 10^8^ stitches would be necessary. This clearly shows the impossibility of measuring a whole artwork with nanometric precision. However, nanometric precision may be required to analyze some parts of a canvas. For example, AFM can visualize physical and visual changes in surface morphology for artists’ acrylic paints and highlight changes caused by exposure to water during wet cleaning [28]. However, these nanoscopic measurements remain a local characterization and the interest is rather to locate the regions of interest to locate the investigated regions at the smallest scales, etc., using the most suitable device described in the previous paragraph. This supposes a first integral measurement of the investigated masterpiece to build an absolute coordinate system to locate the next topographic measurements more precisely. In terms of photography, Sizyakin et al. capture the painting in sections of 20 × 15 cm with a Hasselblad H4D-200MS 50 megapixel camera (with a Charge-coupled device (CCD) sensor of 49.1 × 36.7 mm), equipped with a Hasselblad 120 mm macro lens, resulting in images of 8176 × 6132 pixels. This gives a pixel size of 24 µm. This resolution can be enough to correctly locate the coordinates; however, this remains photography and is barely sufficient to determine location, and not sufficient for multi-scale topographic measurements. Van Hengstum et al. [24] developed a high-resolution topography and color scanner for crack pattern detection in painting measurements using fringe-encoded stereo imaging scanning. This system captures the integrality of a painting at a spatial resolution of 7 microns and a depth accuracy of 34 microns. It takes only 2 h to scan a 39 × 44.5 cm oil painting with 308 stitches, resulting in a topographic map of roughly 97,500 × 111,250 pixels. Callewaert et al. used optical coherence tomography (OCT) for imaging and visualization of Johannes Vermeer’s famous painting *Girl with a Pearl Earring* with micrometer vertical and lateral resolutions, a painting offering over more than five orders of length scales [29]. This apparatus was used by Elkhuizen et al. [21] on the same painting to scan four quadrants of 41 × 41 stitches, covering a total scanned area of 350 × 400 mm with a 8.5 × 67.6 µm pixel area and 3 µm depth resolution, resulting in a 63,636 × 5917 pixel image size and a measuring time of 37 h (we estimate this duration from data in Table 3 of [21]). The technique developed by Van Hengstum et al. is therefore one that allows precise spatial location, coupling color imagery with a topographic measurement. This is relatively coarse in depth resolution for precise microscopic roughness investigations, but covers the art and restoration sectors, allowing an integral measure of the canvas with a highly significant reduction of scanning time. 

**(c)** 
**Create multi-scale topographic processing procedures in order to define a set of morphological descriptors.**


We have seen previously that high-resolution measurements lead to very high informational complexity. We can attempt to answer the following question: what would the resolution be if we would like to discretize our planet with the same number of points measured by the technique developed by Van Hengstum et al. [24]. A simplified calculation gives a roughly estimated elementary paving of $\sqrt{5{.1\;\times \;10}^{14}/\left(94202\;\times \;107487\right)}\;\approx \;200\;m$. Analyzing Vermeer’s 39 × 44.5 cm painting at micrometric spatial scales is equivalent to analyzing the geostatistical information of our planet at a scale of 200 m (120 m if we limit ourselves to land). It is then necessary to describe this landscape, which is strongly multi-scale, using multi-scale topographic analysis. 

There are a multitude of multi-scale methods recognized in the scientific community, as summarized by Brown et al. in 2018 [30]. A generic method for selecting the most efficient decomposition, depending of course on the surface topography, was proposed by Le Goic et al. [31]. However, confronted by this titanic task, it is first necessary to have robust processing tools, validated by the scientific community and having proved their effectiveness. A part of these multi-scale methods for topography are normalized in the ISO standards (ISO 25178 part 2 [32] and 3 [33] with reference to ISO 16610 areal Gaussian filter [34], robust areal Gaussian regression filter [35], spline [36,37], morphology [38,39], wavelet filters [40], and the end effect [41]). 

Concerning the processing of topographic measurements after multi-scale decompositions, a multitude of standardized parameters exist that quantify the shapes of surfaces by taking account of their shape variability (statistical aspect of morphology). These morphological parameters are grouped into five standards (ISO 25178, EUR 15178N, ISO 12781, ASME B46.1, E 16145), each containing taxonomic operational classifications [42]. These standards were created following long debates of surface topography specialists, mainly from the world of manufacturing industry, with a strong inclination towards the mechanical community. From an epistemological point of view, these parameters are intended to control a manufactured product to guarantee, by its surface roughness, its functionality (gloss, tactile, adhesion, friction, heat transfer, biocompatibility, wettability, etc.) and/or to maintain its integrity (wear, corrosion, breakage, etc.). Of these five standards, ISO 25178 is by far the most widely used. It includes almost all of the EUR 15178N standard. The ASME B46.1 and E 16145 standards are almost obsolete in 3D surface finishes. Here we detail the parameter classes of the 3D standard and briefly explain the applicability in the field of artistic painting (Figure A1 in Appendix A shows the numerical values obtained from the topography of a petal in a copy of Vincent Van Gogh’s sunflower (see Section 4) made by our painters). The first category of parameters, which is clearly the most-widely used, describes the "amplitude parameters". It focuses on the amplitude of the roughness, from which the most used parameter is Sa, i.e., the arithmetic means of absolute heights. These parameters make the characterization of the amplitude of the topography of a painting possible: the maximum depth of the valleys and height of peaks, but also the first four moments of the density of probability of the heights, which characterize flatter brushstrokes, thus giving a plateau structure as in that resulting from the technique of the knife art palette. The “spatial parameters”, decorrelated from the amplitude of the paint, characterize, via a study of the autocorrelation function, the orientation of brushstrokes, their degrees of stretching, and their average widths. The "hybrid parameters" characterize the shapes of the structures of the painting, such as the slopes of the roughness, which influence the specular and diffuse rendering of the light on the artwork [43], the fractal aspects (orders) of the shapes left by the brush [44], the average local curvatures that characterize the tribology of the brush-canvas contact [45] and relative to the glossy aspect [46], and the development of the surface, which characterizes the rough aspect of the painting [47]. The “functional parameters” are also amplitude parameters but specifically indicate the distribution of peaks and valleys present in the surface roughness. These are mainly based on a decomposition of the density of probability of the presence of the paint on a plane at a fixed height [48]. The so-called “volume parameters” are tunable and thus allow the characterization of features such as the volume of the cracks in a painting. The “flatness parameters” define the unprocessed canvases [49] of which viewers can gain an impression although it cannot be directly perceived on the painting [50,51]. The last series of roughness parameters is one of the most important in surface topography, namely, “features parameters”. These parameters are derived from a decomposition of the surface topography based on segmentation techniques from Scott [52] and formalized by Wolf [53], using the Wolf pruning algorithm. This method allows significant peaks and valleys to be found. In fact, these segmentation algorithms are intensively used in non-topographic image analysis techniques and many reviews exist for various disciplines [54], including art painting analysis to search for regions of interest [55,56]. Curiously, although these techniques are at the origin of segmentation techniques based on terrain morphology [57], they are more widely used in the field of image analysis where an image is in fact only a complex expression of the topographic gradient (such as a slope derivative). Often, in the image analysis of art paintings, the assumption that the "photographic" image represents a surface topography is too quick a shortcut and is often used without topographic justification. However, one should keep in mind that the reflection of light on a rough surface is a real problem [58]. Therefore, the purpose of this standard is to process topographic surfaces by defining the notions of hills, saddle points, and valleys.

Significant advances have been made in the standard by Blateyron [59], which have been implemented since version 8.0 (2019) of MountainsMap, the reference software in the field of surface processing. The set of all the operators linked to this standard allows multi-scale analysis with filtering method couplings to be performed [60]. The purpose is then to obtain a statistic summary of parameters (area of patterns, radius of curvature, volume, slope, orientation) and a visualization of patterns and trends (Figure A2). The use of this standard to analyze the brushstrokes of a painter is strongly justified by the fact that the calculation procedures of the parameters have been studied [61,62,63,64,65] with a numerical standard surface, allowing the standardized processing procedures to be validated.

The NIST (National Institute of Standards and Technology, Dimensional Metrology Division, Gaithersburg, MD, USA) has created a database to organize information for datasets generated by both measurement and simulation. Machine condition parameters and other process parameters can be stored alongside partial information and roughness parameters computed from the surface profiles [66,67]. In addition, all or part of these topographic methods are implemented in commercially available topographic devices and can therefore be used by the whole community working on the painted work legacy. However, in the case of large topographic maps [24], the complexity of some algorithms can lead to long, not to say unacceptable, computation times [Bigerelle et al., to be published]. To summarize, the standards of surface topography contain a large number of tools to characterize the topographies of the painted works by a joint use (meta algorithm) and then allow the formulation of processing methods directly usable and repeatable by the cultural heritage community.

**(d)** 
**Define a topographic database structure to access the information, regardless of the device used to carry out the measurements and the sets of descriptors.**


It is essential to store information about the morphological content of the investigated surfaces and the measurement conditions. However, the amount of information and its structure remain complex. 

Database structure work has been formalized by Qi et al. [68]. The main problem is to introduce the functional performance of a geometric product. They created an integrated surface texture information system for design, manufacturing, and measurement, called ‘‘CatSurf’’. However, this system is only based on tolerance of manufacturing products [69] and contains the organization of the Geometrical Product Specifications (GPS) standards in surface textures. This means that functionally and integrity of a surface are only described for geometry specifications [70]. However, Berglund et al. proposed a methodology to define the required geometry by results of process simulations [71]. Nonetheless, the data structure proposed by Qin et al. in [69] can be transcribed in the mode of the painting of art by considering, by genericity, that the parts of a painting correspond to parts of a manufactured piece, compatible with all of the GPS standard [72]. For example, significantly fewer works propose a classification of the surface structures in the database, compared to those currently described in computer graphics [73], that will be particularly relevant in surface painting topography to appreciate the painter’s signature.

**(e)** 
**Define, by simulation, the 3D graphic rendering of the surface texture to offer a realistic panel of visual renderings of the surfaces.**


One of the major interests of determining the surface topography is to be able to apply numerical models to simulate physical, or even multi-physical, interactions with the surface. Multi-scale tribological simulations [74] can simulate the contact between the canvas and the brush, rheological models on the paint can explain the shapes obtained on its surface [75], and multi-physical simulations (chemical, mechanical) characterize the degradation of artistic painting (cracks) [76].

A major application of numerical simulation on the topography of a painted surface is the numerical simulation of light reflection on a rough surface. Several distinctions must be made and two categories of models must be differentiated: statistical models and numerical models. 

In statistical models, the topography is often limited to a few very simplistic parameters of electromagnetic considerations (physical models [77]) or optical geometric physically based rendering models (PBRM, [78]) and lead to an estimate of the bidirectional reflectance distribution function (BRDF). In fact, for PBRM models, the BRDF is often computed from images obtained from gonio-spectrometers for the BRDF, whereas the statistical roughness parameters are obtained by the inverse method [79]. The major drawback of this approach is that the roughness parameters are obtained by fitting while making strong morphological assumptions (often Gaussian distributions [80]). It would be more relevant to use the true surface topography and to apply discretized models of surface reflection, whether optical wave [81] or geometrical [82]. It is then possible to reconstruct the Bidirectional Texture Function (BTF) [83] and to obtain a color map derived during the topography measurement in very high resolution. This creates a direct link between the topography and sensory rendering, so that the painter’s technique through his brushstrokes can be better understood. Secondly, another major interest is to simulate the reflection as a function of the position of both the lighting and the observer relative to the painting, thus providing a realistic rendering of the work desired by the painter (better immersion offered to the observer) and not smoothed by statistical models of the BRDF. We can see that storing the topographic image is essential in the conservation of heritage because it is independent of lighting conditions such as photographic measurements. Finally, the knowledge of topography is also of interest in the reproduction of an artwork by additive manufacturing to preserve the topography of the brushstrokes [84]. 

**(f)** 
**Reference the partners (institute, museum, associative world, private collector, etc.) and classify them to build a strategy of elaboration of the most representative database of the diversity of the surface heritage.**


In the Digital Surface Heritage, we propose to define all of the topographic information, among other data, necessary to preserve the heritage and define a realistic computer graphics rendering, in addition to carry out historical and technical investigations. To do this, it is necessary to have cultural information of the painter and his environment. The description of this digital information is part of the “Ontologies for Cultural Heritage” [85]. ISO 21127 standard [86] describes the ontology for the description of data related to tangible and intangible cultural heritage. This standard for interoperability is the result of the standardization work that aims to define the constitutive elements of a domain ontology (classes, properties, definitions, etc.) whose domain is cultural heritage. Regarding ontology itself, this standard also allows the creation of new ontologies (modeling). The enrichment of the CIDOC CRM ontology (International Committee for Documentation of the International Council of Museums, Conceptual Reference Model) [87] continues in parallel with the ISO publication. Regarding the ontology of painted works, OPPRA ontology (Ontology of Paintings and Preservation of Art) [88] is proposed by including sub-ontologies:

CIDOC CRM: this provides the top-level classes and the classes and properties required to capture the provenance information about a painting and its condition, in addition to the conservation/preservation activities that were undertaken;

OreChem [89] was used to model the chemical compounds, chemical reactions and experiments;

OPPRA specific ontologies were developed to describe:-Paints: source (manufacturer/supplier, year, paint product name, identifier, bottle label), paint type, structure, chemical composition, formula, properties, pigment;-Additives: thickeners, stabilizers, preservatives, surfactants, coalescing solvents and defoamers;-Paint degradation: types of degradation (cracking, peeling, fading, discoloration, mold growth), causes (humidity, light, temperature, water, technique), and associated physical/chemical processes/reactions;-Paint analysis methods: macroscopic, microscopic, SEM, TEM, FTIR, infrared, Raman, X-ray diffraction, X-ray spectroscopy (EDS), X-ray fluorescence (XRF), chromatography, synchrotron;-Paint observation: preservation treatments, cleaning, protective coatings, environmental conditions.

We can integrate the OPPRA ontology by adding two sub-ontologies: the GPS ontology described above with some additions for multi-scale aspects, and a topographic imaging ontology that could be created by taking inspiration from the numerous ontologies described in imaging (surface classification [90], image classification in art [91], textures [92], etc.).

**(g)** 
**Implement the set of informatic routines that characterize the relationship between artwork morphology and the parameters described in an information system (concepts and semantic links of ontology relating to painting artworks [88]).**


Once measurements are gathered, morphological indicators are calculated, and databases compiled, it is then necessary to define an information system for data processing. However, because the indicators are numerous, moving towards automation is required, so that the processing becomes more reliable and robust. For this purpose, Najjar et al. [93] proposes to use a technique of over-sampling known as bootstrapping to provide a confidence interval for the estimated roughness parameters. This technique allows surfaces obtained by different categories of processes to be distinguished [94]. These methods have been used to characterize paint damage [95] and thus provide a functional mapping of paint damage mechanisms with their morphological, mechanical, and aesthetic properties [96]. These methods are particularly well suited to investigate quantitative relationships between roughness and one or more physical mechanisms, such as cell adhesion [97], the role of topography on the excessive damage of metals in contact (catastrophic wear) [98], or the role of roughness on the determination of surface mechanical properties [99]. A generic method can then be used to detect the relevant scales of roughness, and to give the characteristic scales and associated normalized roughness parameters that better characterize the multi-scale interaction mechanisms. This allows the identification of the respective influences of external conditions of a multitude of parameters and the determination of the spatial scales in which each parameter influences the topography. This methodology has been applied to successfully highlight characteristic scales of abrasion mechanisms for each associated parameter [100], polymer shaping [101], and impact resistance of a surface subjected to particle impact [102]. 

An ontological approach was proposed by Bigerelle et al. to build the MesRug expert system [103] and was applied in the field of tribology to quantify the multi-scale, multi-criteria, and multi-physical aspects of surface damage [104]. A computation routine was proposed to be integrated in the MountainsMap software (the reference software in surface topography), which allows a simplified ontological approach to be built to classify the roughness parameters by their order of relevance for a set of rough surfaces [105].

### 1.3. A Digital Surface HyperHeritage Example

As part of functional testing of the introduced Digital Surface HyperHeritage approach, the following experiment was conducted:A multi-instrument analysis was undertaken on paintings, made by a panel of ten selected painters reproducing Vincent Van Gogh’s sunflowers;These paintings serve as supports for topographic investigations. Different measuring devices are used (focus variation, interferometry, atomic force microscopy);Surface topographies are investigated and characterized (list of descriptors).

In this paper, we propose the first results of acquisition and archiving of high-resolution digital topography 3D models of selected regions of the artwork. The goal is to characterize the signature imprint of each painter assigned to each object described in the paintings. This digital description is a step forward in the field of painting heritage by offering another vision of an artist’s work. An information system is being built (database), allowing the archiving of the topographic signature of a painting.

## 2. Materials and Methods

The differences in surface roughness characterize a product’s appearance and its tactile features. In the painting process, a complex set of influencing factors form and modify the surface multi-scale topography and its functionality. These are material influencing factors, such as form of the support (canvas or wood, for example), support irregularities (canvas grid), tool characteristics (size of brush, etc.), and paint pigments. The originally created artwork contains individual and authentic topographic information about the painter and his modus operandi. A painter is first and foremost a creator, and they are the basis of the painting process. Their creative process has a significant influence on the appearance and morphology of the painting surface. They are an operator in the painting process and final results mostly depend on their physiological, biomechanical, sensorial, and experience features.

The Digital Surface HyperHeritage approach involves the acquisition and identification of the painting surface topography by capturing all of its functional components at a larger scale, such as centimeter or millimeter (for detecting and fixing the geometry of the canvas), to micro- and nanometers to obtain accurate data on brushstrokes, paint pigments, and other small-sized surface elements (Figure 2).

Indeed, surface topography features many physical responses and information about the surface integrity or functionality. A proper understanding of the full extent of materials and processes used to create a painting would help guide the efforts of art experts to ascertain authenticity, expertise, or authorship, in addition to establish more knowledge about the artist himself.

### 2.1. Art Painting Protocol: “Sunflowers” in Reference to Vincent Van Gogh

#### 2.1.1. Context

The surface of an art painting is a particular case in the study of surface engineering, and its main difference from industrial surfaces is that it is created by the human hand, and not by a machine with previously known process parameter settings. A painting artwork is a complex system that includes many objects or systems of objects. From a sensorial point of view, we can distinguish these objects thanks to the difference in their shapes and colors. For example, in a painting with a representation of a landscape, it is possible to distinguish the sky, the water, grass, trees, and leaves on trees. We can also see houses, their roofs, their windows, their doors, etc. Alternatively, in paintings in which people are depicted, we can distinguish their clothes and their body parts, and we can also look at closer details and consider the “details of details”: fingers, toes, eyes, noses, and lips on a face. It should also be noted that when both observing and creating an artwork, several “generations” of details and forms are considered in the whole picture system. For illustration, we consider the famous series of 7 paintings of “Sunflowers” by Van Gogh (see Figure 3).

Figure 3. shows a series of paintings made by the famous artist Vincent Van Gogh in the period of 1888–1889. First, we see that each painting has an almost identical setting: it is a bouquet of sunflowers in a vase standing on the table in front of a monotone background. Furthermore, it is possible to distinguish each flower in the bouquet: this is the second generation of objects. Each flower has petals, leaves, and a sunflower head, which represent the third generation of objects. The sunflower head has grains, and to draw a petal, the artist may perform several actions, because the base and end of the petal are performed with two different movements or using different brushes, resulting in a fourth generation of objects. This form of analysis applies to any painting artwork.

Thus, a painting artwork is a complex system that includes a set of multiple objects reproduced by the artist in different ways and conditions. Therefore, physically, these objects have different shapes, sizes, positions, and surface morphology. In addition, there are many factors that affect the formation of a painted surface, such as the materials and tools used, the painter’s dexterity, the techniques and styles, and environmental conditions. From a surface engineering perspective, the surface of the art painting is an integral set of heterogeneous, multi-scale, and spatially stochastic surfaces. 

Therefore, it is necessary to implement a three-dimensional multi-resolution and multi-technique approach in the measurement strategy to capture all of the features and particularities of the paint work. 

Artists are known to make reproductions of their own or other’s paintings, with the aim of recreating visually similar copies: this represents several versions of paintings with the same settings. The series of Van Gogh’s sunflowers includes three versions of the painting "Vase with fifteen sunflowers" (Figure 3c–e) and two versions of the painting “Vase with twelve sunflowers” (Figure 3f,g).

In both cases, all versions of the paintings seem to be maximally similar, however, even upon visual inspection, it can be concluded that identical objects from each version differ not only in color, but also in size, shape, and position. When creating a repeated version of a painting, the artist tries consciously to make changes to the interpretation of the settings, or vice versa, trying to reproduce in detail a visually close version. Regardless, the author never recreates an absolutely identical surface of the painting. The same is true when trying to copy other artists. From a surface engineering perspective, the surface of each artistic painting is heterogeneous and unique, like human fingerprints.

When studying the influence of biomechanical factors on the created surface of a painting, it is necessary to compare the topographic signatures of different artists to establish the differences between them. This is especially important for obtaining fundamental models to apply in authentication and expertise studies.

In reality, we deal with complex paintings and not elementary brush strokes. In this case, comparing topographic signatures of a group of painters requires minimizing the risk of differences in their interpretations of shapes, sizes, and positions of objects. Each artist should perform movements in the most habitual and natural way. However, for a correct comparison of objects of an identical painting, it is necessary to prepare the experiment in such a way that all participants unconsciously reproduce the sizes and shapes of the figures by maintaining the same spatial resolution.

#### 2.1.2. Selection of Painters and Painting Instruction

In the case of a complete painting, to be able to statistically compare topographic signatures of different artists, it is necessary to ensure the most similar conditions for the performance of the artwork, in which the author freely creates his painting. It is necessary to exclude as much as possible factors that contribute to the addition of variables for analysis. All artists must reproduce the same picture and must use the same materials and tools. It is also important to ensure that each artist materializes his interpretation of forms, transferring the sizes and positions of figures from a given image as accurately as possible.

To digitize, analyze, and compare topographies containing artists’ signatures depending on the given settings (i.e., objects to reproduce), the following experiment was performed. Ten selected painters (2 professors and 8 students) from the art history department of the university were proposed to reproduce a sunflower based on Van Gogh’s painting “Vase with twelve Sunflowers” (Arles, August 1888; Neue Pinakothek, Munich, Germany). The dimensions of the real artwork are 91 × 72 cm. For reproductions, a sub-region was selected that includes one sunflower and corresponds to the real dimensions of 20 × 20 cm (see Figure 4). However, to avoid participants’ attempts to accurately represent the sunflower of the Van Gogh painting, a photo of a real sunflower similar in appearance and shape to a sunflower from the Van Gogh painting was found and printed in dimensions 20 × 20 cm (see Figure 4, right side). 

The idea of the experiment is based on the fact that each participant reproduces the sunflower from the photo on a canvas of an identical size. This fragment was chosen for representation for several reasons:-the presence of different types of objects for reproduction (sky, petals, sunflower head, grass);-the shape of the flower allows several identical objects from different angles (petals) to be viewed;-painter repeatability: each painter is able to reproduce objects, accurately maintaining their forms, sizes, directions, and positions.

The main idea is to finally obtain several artworks of different painters, but maintaining the same spatial resolution to allow comparison of the similar objects.

#### 2.1.3. Canvas and Paintings

Each painter obtained a kit including:-1 printed picture of a sunflower with dimensions 20 × 20 cm;-1 pre-coated linen canvas with dimensions 20 × 20 cm;-1 set of oil paints “Pebeo XL” (20 colors);-1 round brush “Gerstaecker” with synthetic white fibers, size 10;-1 bottle of turpentine “Lefranc Bourgeois”;-1 list of instructions and tutorial video.

Each participant received a personal kit of materials for work and individually performed the painting at home or in a studio at the university. According to the results of the questionnaires, the duration of the painting’s performance was from 2 up to 20 h. It took roughly 3–4 weeks for curing and drying of the paint before the object was able to be measured. Two painters were proposed to perform a second version of the painting to obtain samples for investigation of “intra-painter” topographic signature similarities. Finally, thirteen finished paintings with panel dimensions of 20 *×* 20 cm were produced. These paintings serve as supports for the following topographic investigations. The appearance of one painting from each participant is shown in Figure 5.

### 2.2. Multi-Instrumental Strategy

#### Selection of Metrology Devices

There is an increasing use of techniques to study cultural heritage objects depending on the subject of the study (see Section 1). These nondestructive, contactless, optical techniques provide data regarding large surfaces, and sometimes even the full object. The purpose of our research project first requires measurement of the geometric and spectral properties of the art using modern measurement methods. Novel measurement systems perform well regarding the 3D capture of depth information by combining both colorimetric and topographical data. Depending on the apparatus, it is possible to cover a large range of lateral resolutions to capture the topography of artistic painting surfaces at the scale containing the elementary information of sensorial rendering, in addition to fine details such as cracking patterns throughout a whole painting.

For the investigation of surface topography, a focus variation microscope (Alicona^TM^, InfiniteFocus G5, Raaba, Austria) with a 10× objective lens was used. A focus variation metrological system combines the shallow depth of field of an optical system and its vertical scanning to provide both true color and topographic information at the same time (that also allows image analysis to be performed) from the variation in image sharpness on the sensor. The visual correlation between the optical color image of the partial surface and the height information, which are often related to each other, is therefore essential to analyze painting surfaces with high-resolution scales from centimeters to micrometers [106,107].

The advantage of this technique is that it offers topographic maps alongside images of surfaces up to 10 × 10 cm (using stitching) with a micrometric resolution. However, at the present time, this technique does not allow measurements with nanometric and sub-micrometric accuracy in the required locations.

Sub-micrometer and nanometer resolutions were achieved using interferometry and atomic force microscopy. However, the acquisition time increases, because each tile only captures a very small area of the painting. Nonetheless, these techniques allow the measurement of surface topography at desired locations of interest, such as micro cracks, with the high accuracy required for analysis.

A scanning White-Light Interferometer (SWLI) (Zygo^TM^, NewView 7300, Middlefield, CT, USA) with magnification of 50× was chosen to capture topography with sub-micrometric accuracy.

Atomic force microscopy (Bruker^TM^, Dimension Edge, Santa Barbara, CA, USA) was used to capture topography with nanometer accuracy.

An overview of the above-mentioned techniques used and their specifications can be found in Table 1.

### 2.3. Measurement Strategy

Surface topographies of 10 paintings were measured and characterized. A surface including the region of a petal (see Figure 6 of one of painting is used as standard test surface in order to show topographic maps and results of the subsequent analysis.

For investigation of surface topography containing objects of micrometer, millimeter, and centimeter scales, a focus variation microscope (Alicona^TM^, InfiniteFocus G5, Raaba, Austria) with a 10× objective was used. The lateral resolution aims to determine how many pixels are used to resolve local heights. Increasing it leads to higher details, such as brush strokes. Vertical resolution changes the image stacking: the more images are added to the stack, the more likely that a point of maximum contrast in association with the field depth is detected [108].

For the painted surface, we used 1.76 µm for lateral resolution and 100 nm for vertical resolution. This provides a good compromise between measurement time and roughness accuracy (Figure 6).

Finally, a topographic map was obtained, with dimensions of 81 × 21 mm (46,022 × 11,932 points) by stitching 56 × 14 (784) elementary surfaces. Measurement took 9 h (with vertical scanning speed from 1000 to 3000 µm/s and measurement speed ≤1.7 million measurement points/s). Each measurement set contains a surface topography file (*.al3D), an image (*.bmp), and a quality map (*.bmp). Each set including topography, image map, and quality map has a size of roughly 10 GB. Each file including topography layer (*.sur) has a size of roughly 2 GB (depending on dimensions of the petal).

The following surfaces were measured on each of the available paintings: -A surface including a region of the petal with angle of 0° compared to the lateral axis. The dimensions of the measurement region were defined to fully contain a flower petal (see Figure 7, zone 1) with dimensions of 50 × 25 mm (depending on the petal).-A surface with dimensions of 10 × 10 mm (see Figure 7, zone 2)-A surface with dimensions of 1 × 1 mm (see Figure 7, zone 3).

Then, a surface with dimensions of 100 × 100 µm was measured on each available painting using a white light interferometer (Zygo^TM^, NewView 7300, Middlefield, OH, USA) with magnification 50× (see Figure 7, zone 4). For determination and digitization of a nanometric object, a sub-region of 1 × 1 µm was measured on each painting using atomic force microscopy (Bruker^TM^, Dimension Edge, Santa Barbara, CA, USA).

## 3. Results and Discussion

### 3.1. Multi-Scale Topographic Map

Surface topographies of the selected regions were obtained. First, topography measured by focus variation microscopy is presented in Figure 8. The result represents a digital three-dimensional model of the surface topography containing a flower petal measured with micrometric accuracy. At the top of the figure, the model is presented as a color model, and at the bottom of the figure as a topographic map (pseudo-color corresponds to height).

Topographic models obtained as part of an experiment for previously presented surface areas are shown in Figure 9, Figure 10, Figure 11, Figure 12 and Figure 13. Depending on the selected scale of the surface topography, various functional objects of the painting can be observed:-Centimeter scale (Figure 7, zone 1). Brushstrokes and geometry of canvas are observable. We can quantify the influence of the brush size and shape (Figure 9);

-Millimeter scale (Figure 7, zone 2) shows precisely the imprints of brush hairs. Size and form of the traces depends on type, size, and density of the brush hairs and the type of paint (Figure 10);

-Micrometer scale (Figure 7, zone 3) can possibly represent either small-sized damage, such as microcracks and bubbles, a paint mixing border, and other micrometric sized objects (Figure 11);

-Sub-micrometer scale (Figure 7, zone 4) allows the microstructure of paint coating formed by chemical and physical processes during drying and film formation to be observed (Figure 12);

-Nanometer scale (Figure 7, zone 5) determines and digitizes paint pigments (Figure 13).

### 3.2. Multi-Scale Analyses

A multi-scale analysis strategy was implemented on surface topographies of 10 paintings (see Figure 5). A surface including the region of a petal (see Figure 7) of one painting was used as a standard test surface for illustration of a multi-scale decomposition to demonstrate the range scales that be used to define painting surface contains.

#### 3.2.1. Multi-Scale Decomposition 

To determine the most relevant scale for painting signature comparison in regard to painter aptitude, multi-scale decompositions of the surfaces were conducted. The topographies were first analyzed using the MountainsMap^®^ software package from the raw measurements. By applying a high pass (HP) filter, we removed the roughness scales above a defined threshold and obtained only the lowest roughness scales (Figure 14).

#### 3.2.2. Multi-Scale Topographical Graph 

By sequentially increasing or decreasing the threshold value, we revealed the spectrum of all of the topographic scales included in each measured surface, namely the “surface roughness”, with the HP filter. Then, we calculated for each surface of the spectrum a large number of topographic parameters. We then obtained for each of these parameters its evolution depending on the filter cut-off and, by extension, depending on the scale (Figure 15).

### 3.3. Discussion

First, we analyze the multi-scale graph of Sa as a function of scale (Figure 15). It clearly shows four regions that qualified as stages, ranging from a few micrometers to one centimeter. The spatial resolution of the AFM, the focus variation microscope, and the interferometer means that the three techniques only have an area of about 10 × 10 microns in common. Indeed, the different selected devices only share a common part over a short length. Stage 1 corresponds to a range from 1 to 15 micrometers. The evolution of Sa as a function of the filter cut-off follows a linear scaling law (log-log). It has been shown [109] that the presence of a linear relationship characterizes a power law, whose slope in a log-log scale represents the fractal dimension (the slope of the line is equal to 3-D where D is the fractal dimension). This highlights that the amplitude of the roughness follows a scaling law that depends on the type of device. The scaling law of the focus variation microscopy clearly shows a lower roughness than those of both the AFM and the interferometer, and a smaller fractal dimension. We can hypothesize that these three techniques capture the fractal aspect of the roughness, but the transfer function of each device [19] under small scales generates a change in behavior (rougher for the interferometer and smoother for the focus variation). A remarkable fact is that these three lines, in log-log scale, concur at the same point (spatial scale of 15 micrometers) and this point constitutes the end of stage 1. We can make the hypothesis, confirmed by the analysis of the topographic maps (see Figure 14, scale 5–15 µm), that it constitutes the scale of the pigments measured on the canvas itself, i.e., after impregnation and evaporation of the oil. From 15 to 70 microns, a linear regime in log-log scale also appears, whose amplitude values measured by interferometry and focus variation are almost identical. Topographic maps clearly show that, at these scales, pigment clusters begin to appear (Figure 14, scale 40 µm), attributable to a non-homogeneous dispersion of the pigments in oil. From 70 to 600 microns, a linear log-log slope characterizes the multi-scale (fractal) morphology of the brushstrokes. This manifestation of a power law highlights the fractal aspect in this range of brushstrokes, representing the various traces left by some gatherings of different sizes of brush hairs (Figure 14, scale 300 µm).

From 600 microns (Figure 14, scales 600, 2000 µm), the log-log evolution of Sa with the cut-off becomes weak, or just a slightly increasing stagnation of the amplitude of roughness. At these scales, called stage 4, the morphology of the brushstrokes begins to form shapes that will constitute the object desired by the painter. To corroborate the interpretations made previously, other standardized roughness parameters are described, detected as relevant by the bootstrap-type analyses described above. First, we examine the parameter Spk, which represents the amplitude of the peaks protruding from the roughness [48]. The analysis of the multi-scale graph indicates the presence of a strong noise for the focus variation microscope for values below 15 microns. This appears to show the difficulty to accurately measure by optical focus the sharp peaks of roughness. Above 15 microns, both optical devices have a linear scaling law (log-log) showing the ability to describe the fractal aspects of the most important peaks. However, focus variation microscopy amplifies the amplitude of the peaks compared to interferometry, which also amplifies it compared to AFM. We can therefore observe that optical techniques have a tendency to amplify the amplitude of the highest peaks of roughness. This phenomenon is certainly due to a smoothing imposed by the focus variation algorithm (contrast function) and, to a lesser extent, to the focusing techniques in interferometry. 

The Str parameter characterizes the directionality of the paint texture [110]. One of the advantages of plotting this parameter as a function of scale is to determine at which scales the structures are more or less directed. A Str value close to unity implies a purely isotropic structure, in which no direction appears on the painted canvas. In contrast, a Str value close to 0 would show unidirectional shapes (rectilinear brushstrokes). The analysis of the Str parameter as a function of scale shows without any ambiguity a scale transition for both optical devices at 80 microns. Under this spatial scale, the structures appear rather isotropic. Above this scale, the structures appear strongly anisotropic. Note that this transition scale corresponds to the beginning of stage 3 previously shown by the Sa multi-scale graph. Below 80 microns, the roughness is made up of peaks and unstretched valleys, which appears to correspond to direction-independent morphologies and thus confirms a regime of aggregated pigments. Above 80 microns, the directionality imposed by the brushstrokes begins to appear.

The smoothing effect imposed by the focus variation is strongly represented by the parameter of roughness slopes (Sdq). For AFM and interferometric microscopy, remarkably, the slopes are constant and the scales investigated are identical. These apparatuses thus seem relevant to characterize the local facets in a global integration of the surface morphology. Knowing that the orientation of these facets is one of the most important parameters for the reflection of light on a rough surface (Torrance–Sparrow, Ward, Cook–Torrance models, etc. [111]), the interferometer will be particularly suited to simulate, according to the topography of the rough surface, computer graphics that render integration of the smaller scales of a painted picture (see Figure A3).

The curvature of the surface peaks can be characterized by their local radius of curvature. If this value clearly allows differentiation of the pigment area and its shape (respectively 80 and 600 microns), the values obtained by the three measurement techniques show differences of a factor of 100 between AFM and focus variation microscopy for small scales, and a factor of 10 for medium scales between the interferometer and focus variation microscopy. It is an established fact [112] that the measurement of the curvature of roughness peaks is highly complex and dependent on the experimental measurement techniques. To calculate this value for each roughness pattern it is necessary to approximate the roughness peak by a sphere in the least square sense, and a simple blurring effect caused by an experimental technique can drastically amplify the radius of curvature of the roughness peaks. 

## 4. Conclusions

In recent years, there has been an increasing interest in scientific investigations of cultural heritage to build new resources from the perspective of preservation, development, experience, and transmission of all aspects of cultural heritage. In this paper, we propose the Digital Surface HyperHeritage approach, which takes a step forward in the digital artistic painting heritage description. This approach allows identification of the topography of painting surfaces at scales containing all of its functional components, from large to small-sized surface elements. The first 3D topographic digital description of five selected surface sub-regions of the artwork were obtained using different measurement techniques—focus variation microscopy, atomic force microscopy, and interferometry. In summary, these results show the ability to acquire and digitize a painting surface with high resolution, including the capture of objects of various sizes from centimetric to nanometric scales.

However, some strategy must be developed to integrate the standard roughness and measurement conditions and to link them to the art ontology. The main difficulty is to define a global absolute reference of paintings that is sufficiently precise to carry out increasingly local measurements on objects representative of the paintings, and such that no topographic scale break occurs. This certainly requires a multi-instrument topographic measurement. The process outlined in this paper highlighted a number of questions that require further investigation and development. Finally, to develop a strategy of elaboration of the most representative global database of the diversity of the surface heritage, it is necessary to reference and classify the cultural institutes, museums, and associations, and to provide them with a computer application using our ontologies.

## Figures and Tables

**Figure 1 sensors-20-06269-f001:**
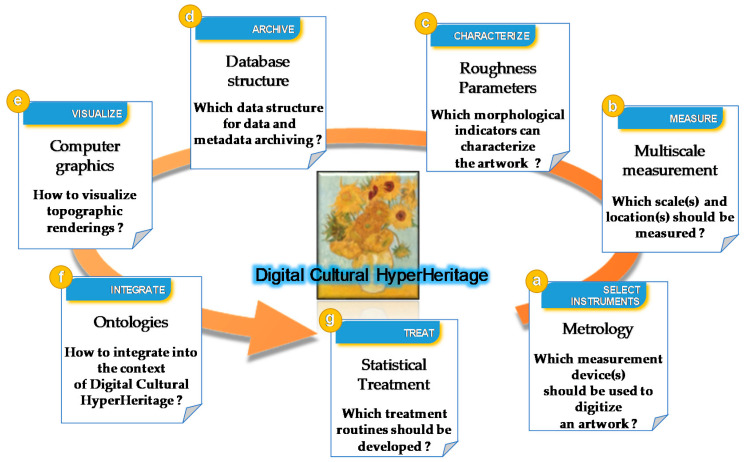
Conception of the Digital Surface HyperHeritage approach.

**Figure 2 sensors-20-06269-f002:**
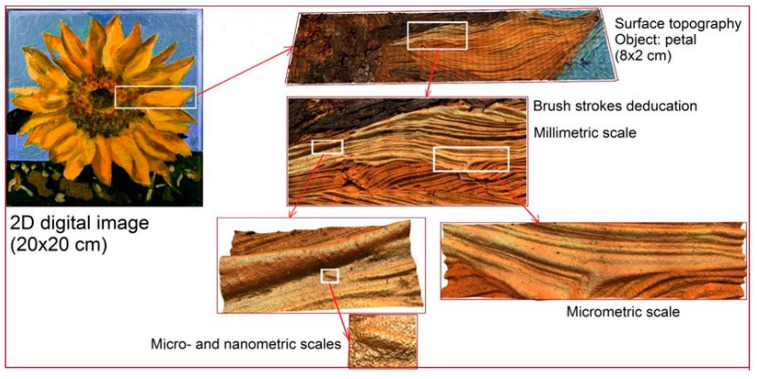
Schematic summary of scales contained in the studied artwork.

**Figure 3 sensors-20-06269-f003:**
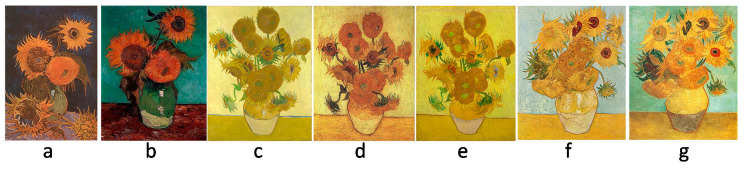
Series of 7 paintings of “Sunflowers” by Vincent Van Gogh: (**a**) “Vase with five sunflowers”, (**b**) “Vase with Three Sunflowers”, (**c**) “Vase with fifteen Sunflowers”(Arles, August 1888), (**d**) “Vase with fifteen Sunflowers (repetition)”, (**e**) “Vase with fifteen Sunflowers (repetition)”, (**f**) “Vase with twelve Sunflowers”, (**g**) “Vase with twelve Sunflowers(repetition)”.

**Figure 4 sensors-20-06269-f004:**
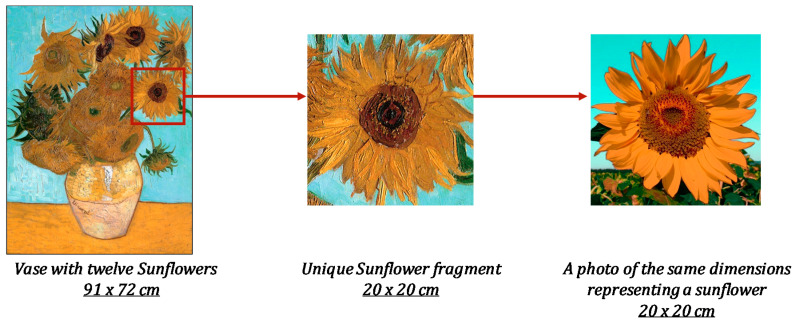
Schematic representation of the selection process of a subject for reproduction.

**Figure 5 sensors-20-06269-f005:**
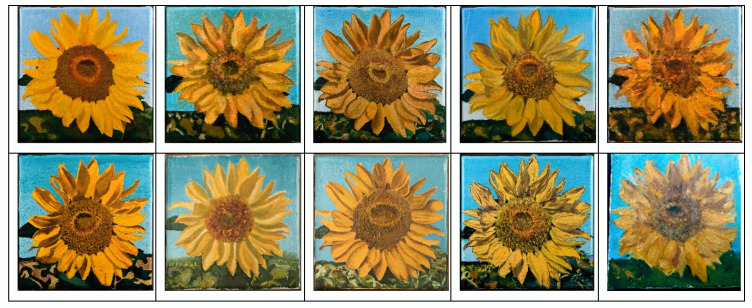
Paintings for topographical investigation.

**Figure 6 sensors-20-06269-f006:**
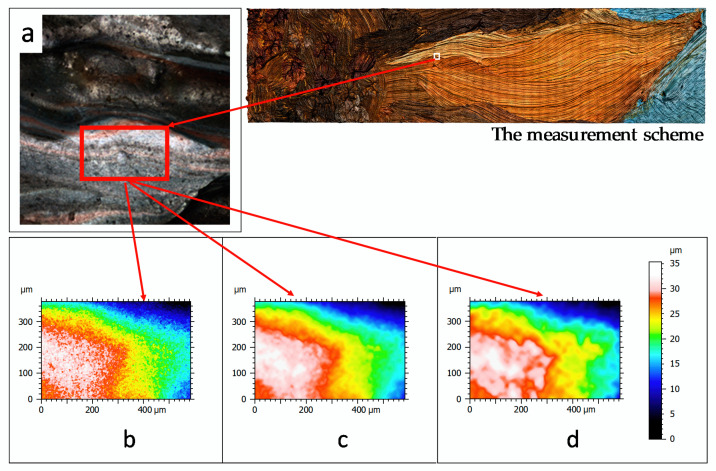
Height map visualization of a painting (**a**) using 10× magnification, coaxial and ring lights with a vertical resolution of 100 nm and a lateral resolution of 1.76 µm (**b**), 3 µm (**c**) and 10 µm (**d**).

**Figure 7 sensors-20-06269-f007:**
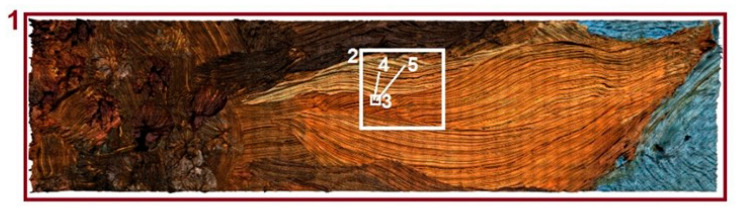
The measurement scheme.

**Figure 8 sensors-20-06269-f008:**
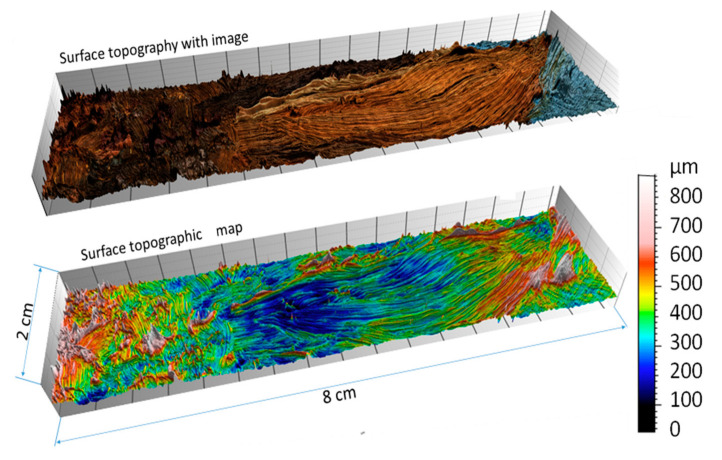
Digital model of surface topography using focus variation microscopy.

**Figure 9 sensors-20-06269-f009:**
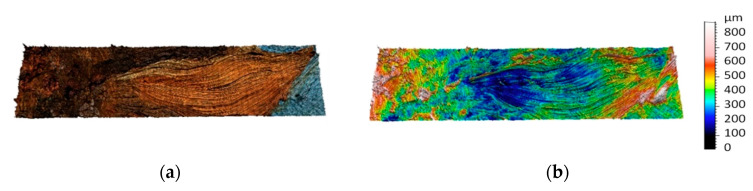
Topographic digital model of a centimeter scale region (8 × 2 cm)—focus variation measurements: (**a**) surface topography with image and (**b**) surface topographic map.

**Figure 10 sensors-20-06269-f010:**
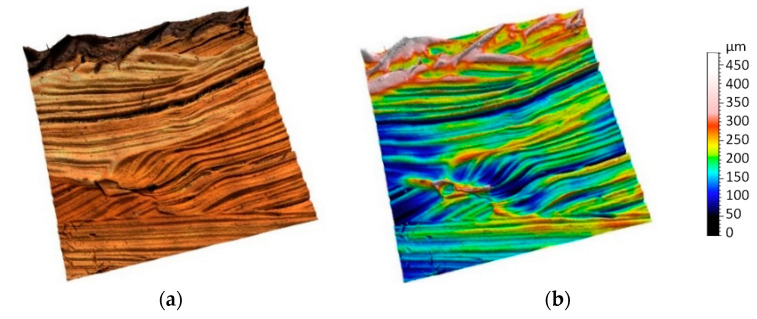
Topographic digital model of a millimeter scale region (10 × 10 mm)—focus variation measurements: (**a**) surface topography with image and (**b**) surface topographic map.

**Figure 11 sensors-20-06269-f011:**
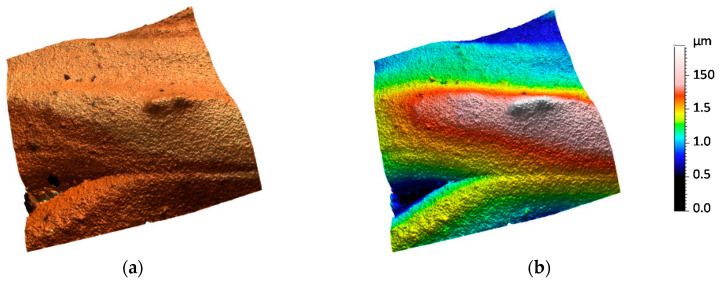
Topographic digital model of a micrometer scale region (1 × 1 mm)—interferometric measurements: (**a**) surface topography with image and (**b**) surface topographic map.

**Figure 12 sensors-20-06269-f012:**
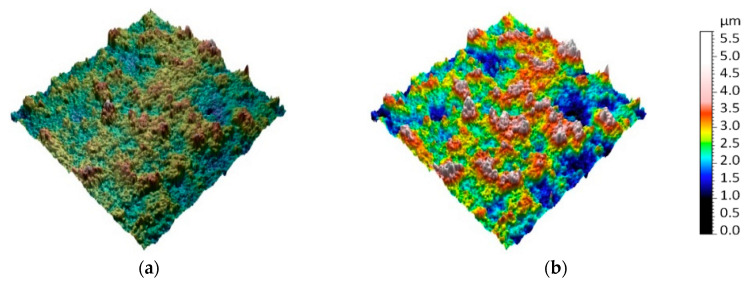
Topographic digital model of a sub-micrometer scale region (100 × 100 μm)—interferometric measurements: (**a**) surface topography with image and (**b**) surface topographic map.

**Figure 13 sensors-20-06269-f013:**
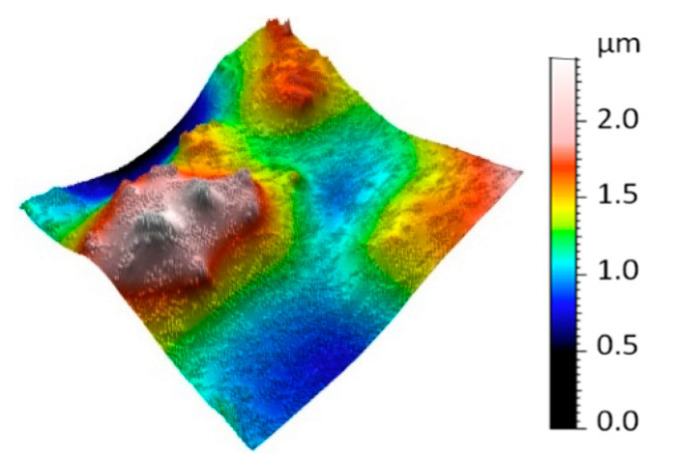
Topographic digital model of a nanometric scale region (10 by 10 µm) measured by atomic force microscopy.

**Figure 14 sensors-20-06269-f014:**
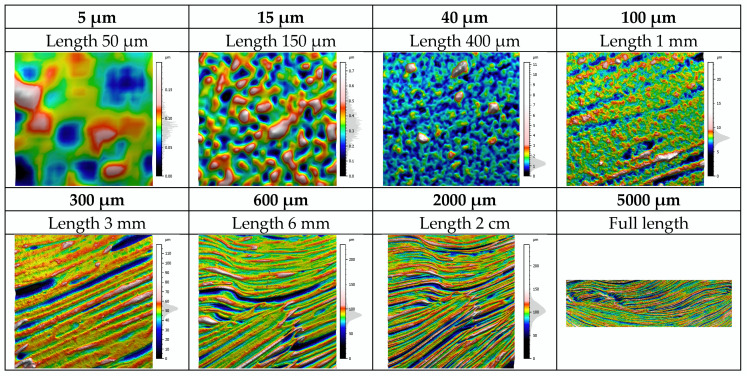
Principle of surface filtering (examples given with a threshold of 5, 15, 40, 100, 300, 600, 2000, and 5000 µm using a high pass filter). The full-length scale corresponds to the “optimal” filtering to appreciate the topography describing the artistic painting of all objects present on the canvas, independently of the brush stroke signature.

**Figure 15 sensors-20-06269-f015:**
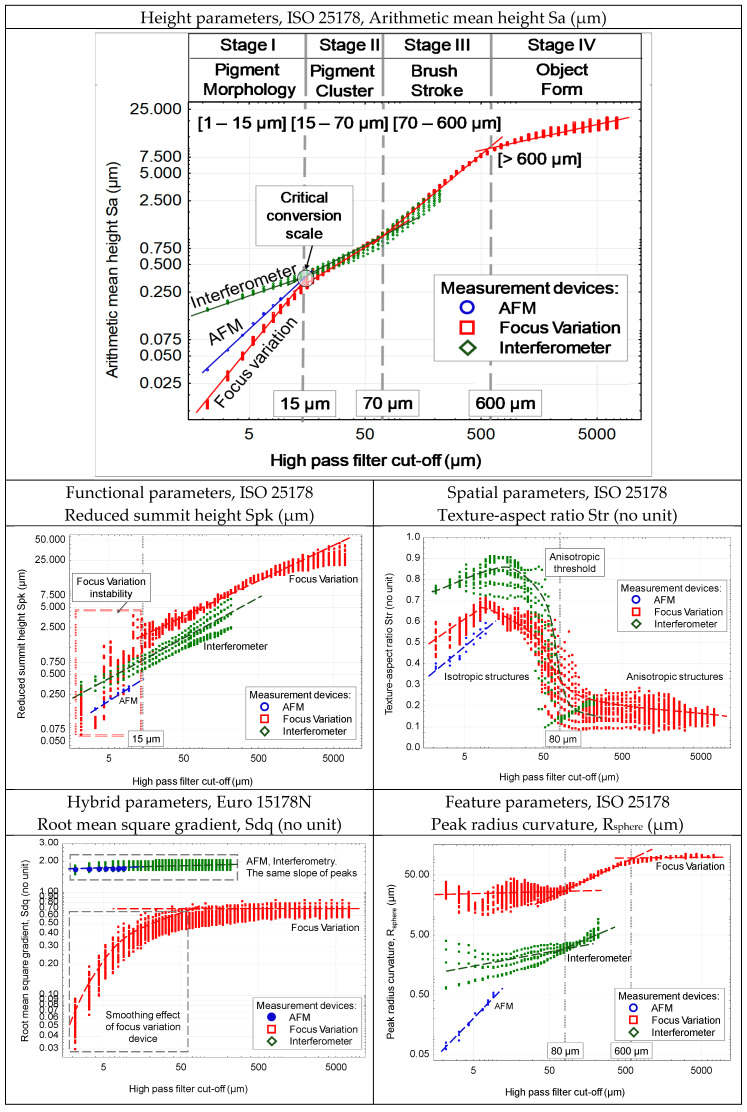
Multi-scale and multi-parameter characterization of the petal painting, from referenced to the Van Gogh “Sunflower” painting (see Figure A1). Sa, Spk, Str, Sdq, and radius curvature of asperities are plotted as a function of the scale (cut-off of Gaussian high pass filters).

**Table 1 sensors-20-06269-t001:** Summary table of surface topography investigation techniques.

Technique	Specification	Vertical Resolution	Lateral Resolution	Field of View	Phenomenon to Study
Focus variation Magnification: 10×	Allows topographic maps to be obtained combined with color images of surfaces up to 10 cm by 10 cm with high resolution	100 nm	1.76 µm	1.62 × 1.62 mm	Geometry of canvas, brushstrokes, painter’s modus operandi.
Interferometry Magnification 50×	Allows acquisition of topography with micrometer and sub-micrometer accuracy, less influenced by color, which allows certain mistakes during the measurement process to be avoided	10 nm (using motorized extended scan)	0.52 µm (with Sparrow criteria)	0.14 × 0.11 mm	Traces of brush hair, small-sized damages, bubbles, pigment clusters
Atomic Force Microscopy (peak force tapping mode)	Allows topographies with sub-micrometer and nanometer accuracy to be acquired	0.2 nm	0.5 nm	10 μm	Paint pigments

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
