# Peer review of "Digital Cultural Heritage Preservation in Art Painting: A Surface Roughness Approach to the Brush Strokes"

_sensors, 2020, doi:10.3390/s20216269_

Round 1
Reviewer 1 Report
The paper presents an interesting approach about the use of metrology models applied to study and reproduce paintings , including the development of a database for archiving and sharing the topographic signature of a painting.
I’m convinced that the proposed approach fits with the aims of the journal that encourages theoretical or experimental dissertation about signal processing and data fusion.
Nevertheless, the paper shows a more lack of scientific data and definitions about used resources and technologies and analysis tools applied to architectural heritage.
Generally speaking, the paper organization is quite confused. I suggest to the authors to adopt the canonical scientific structure, for example a paragraph of related works or state of art is mandatory. In the current version the choice of some titles in the current version is misleading. For example the 2. Heritage of the art painting seems a motivation of the work, highlighting also some achievement, but merging also state of art (in fact it contains the majority of the references). It is particularly difficult to understood the goal of the chapter "Sensorial perception transmission".
Under the chapter Technical heritage, the paragraph 3.1. Individualization, authentication, falsification and artist’s working methodology understanding is far to be technical, but it assumes a discursive and shallow style.
The paragraph of the methodology is usually much more consistent and with steps of integration between the phases, in this case the methodology consists in the simple application of different sensors to gradually smaller portions of the analyzed surface.
Nevertheless, it is interesting the application of analytic tools. The idea to match morphological investigations (Focus Variation) with qualitative ones (Interferometry, Atomic Force Microscop) is a practice in the field of cultural heritage, but I find original and innovative the idea of developing a morphological model to be exploited as an archive starting from that kind of data. Other workflows, that is worth to mention usually apply such kind of data collection and/or enrichment to models derived by laser scanning techniques or photogrammetry.
It is not clear if the definition of Hyper Heritage is by authors or if they only embrace this paradigm adding the term Digital and enlarging the concept to new ICT application. If this last is the correct situation, the literature referring to Hyperheritage have to be cited and a brief discussion on this concept and its history could be interesting.
Line 37-39
The authors state “But for the present, the analysis tools and methods used are not still introduced into the research as techniques for digitization and preservation of the world art painting heritage.”, although digital investigation of paintings and their 3d or topographic mapping are low investigated field some works are available and the authors need to cite them.
Moving towards the conclusion, I can affirm that at least main achievements and future directions are mandatory. In this current version the discussion and conclusions are treated at a very high level of abstraction and are not sufficient for a scientific paper. In particular the discussion is mainly based on a simple visual comparison between RGB and false color deviation map that shows interesting results but needs to be deepened.

Reviewer 2 Report
The paper describes a multiscale analysis of the surface of paintings. Indeed, paintings are not flat, as paint accumulates creating a 3D topography. Authors describe very well the scenario of 3D analysis of paint surfaces for applications like conservation, communication and attribution of paintings. However, the contribution seems not very mature, as very preliminary results are provided, and a discussion or a concrete plan for going forward is missing.
For this reason, I suggest to clarify what the present contribution tackles in particular, even if within the frame of a wider project. I guess the Digital Surface Hyper Heritage project is still ongoing, but without a precise frame, the reader does not capture the main focus of the work, which touches many aspects but does not go in depth in any of the following directions: is the focus on the acquisition issue? On the modeling and simulation of the paint surface? On the analysis of the reconstructed model and the identification of proper descriptors to get a painting signature? On the definition of a proper database structure?
A few comments in more detail.
I really enjoyed reading sections 1 and 2, they give a very interesting description of the scenario. One issue: the introduction describes three experimental phases, but #1 and #3 are not addressed later on. The following line explains this contribution reports the first results in acquisition and archiving; but then, I would expect more technical insights on the acquisition process (instead, it seems applying state of the art device and methodology was sufficient) and actually there is no result about the archiving in section 3.3.
There is no explicit section on the state of the art. I don't mind, unless the novelty of the contribution with respect to previous work is clear. I think that similar works in the field of computer graphics could be discussed. For instance, there are a few works on the 3d scan and analysis of paintings (especially for crack identification) in the proceedings of the EG workshop on Graphics and Cultural Heritage (e.g., "Development of a high resolution topography and color scanner to capture crack patterns of paintings, by van Hengstum et al., in GCH 2018).
In general, section titles are a bit verbose, they make me loose the focus. Title "3. Technical Heritage" is too vague - is this a more technical section, with implemantation details? It seems not - conversely, the discourse here is still at introductory level (e.g., the paragraphs at lines 195, 199, 222) and shows also some redundancy (e.g., the paragraphs at lines 210 and 215). I suggest to summarise this section. The scheme in s
Figure 1 is very interesting; I would like to know if it is based on some reference taxonomy. I guess this will help deduction of the painter style in the long term, but the current results are very far from this goal.
Section 4 reports the experiment. It is really interesting to see the results of the different scale acquisitions. Did you analyse how the resulting height changes for the "same" position through the scales? The results analysis remains unfortunately at qualitative level, but some more discussion about the potential of shape analysis on the models to understand further insights would be an added value. I did not get the meaning of figure 9 - what does determination and digitization of paint pigments mean? do we get an information about the chemistry of the paint?
To conclude, assuming this is just a very preliminary step in the framework of an ongoing project, I strongly suggest to add concrete steps foreseen to continue the work concerning the experiment (e.g., evaluate differences among the painting of the ten artists) and towards the goals given in the introduction (e.g., shape descriptors suitable to provide a significant signature of a painting / painter).
Reviewer 3 Report
This manuscript titled "Digital Cultural Heritage Preservation in Art Painting. A Surface Roughness Approach of the Brush Strokes" proposed Digital Surface HyperHeritage approach to understand painting heritage. The authors showed the preliminary experimental results of acquisition and archiving of a high-resolution digital topography 3D model of a selected area of the artwork. This approach seems very useful in understanding the characteristics of painting heritage, and the results of the experiment are also interesting. Overall, this manuscript is assessed to conform to the purpose of the special issue of Sensors, “Sensors for Cultural Heritage Monitoring”. However, if some additional explanations are included, it will help readers understand the paper.
- The authors selected three measuring devices (Focus Variation, Interferometry, Atomic Force Microscopy) for topographical investigations of the artworks. Then it seems necessary to explain why these devices were chosen through comparing with other techniques, such as 3D-scanning.
- Further explanation is needed as to how the measurement results of micro-area could be linked to the macroscopic characteristics (such as brush strokes) of the whole artwork. In particular, the fine-structures presented will be transformed over time, depending on the characteristics of the artwork. Hence, further discussion would be needed about usability.
Round 2
Reviewer 1 Report
The present version of the paper seems significantly improved.
So I suggest as it is.
Author Response
Thanks for all your remarks that have increase the quality of our paper
Reviewer 2 Report
The authors did a very hard work on the paper. I am really impressed by the effort put in the revision, but I have to say not all the changes lead to an improvement from my point of view.
I cannot see other reviewers' comments, so I guess other reviewers asked for changes different from what I felt important. I will give my opinion on this version and I guess finding a balance between different requirements will be in the hands of the editor.
In the previous version, I appreciated the introduction, while I found the results section a bit too short and unbalanced. Now authors added more technical details and a discussion section, so I also understood and appreciate more the value and novelty of the method.
On the other side, I also already pointed out a lack of focus of the current contribution with respect to a wider research project (the Digital Surface HyperHeritage). This issue is now painfully worse.
I really respect authors' effort, which is amazing, but let me speak frankly. 8 pages of introduction including 100+ references, plus section 2 about all the standards and ontology one could think of, is crazy in my opinion. All this content has value and interest, but is only partially related with the core contribution, which is, in my opinion, the work of digitization performed at different scales, with different instruments, and the effort of representing and characterizing in a multiscale space the painting. I would even include the appendice content within the paper, maybe. But this is the core, the subject. The rest is context. So, I am happy to know about the framing of the work, to get motivations and relations with a bigger picture, but the former introduction, even if with some unclear points, was good enough!
Still, I don't see how the ten different paintings by different artists were taken into account in the experiment, but I maybe lost something (the paper is now really long).
Overall, I would suggest to summarize the first 13 pages to-say- 2 or 3 max pages, reporting only the facts that really comes into play in the core contribution.
